# Optofluidic memory and self-induced nonlinear optical phase change for reservoir computing in silicon photonics

Chengkuan Gao [1], Prabhav Gaur [1], Dhaifallah Almutairi[1,2], Shimon Rubin [1] ✉ & Yeshaiahu Fainman[1]

Nanophotonics allows to employ light-matter interaction to induce nonlinear optical effects and realize non-conventional memory and computation capabilities, however to date, light-liquid interaction was not considered as a potential mechanism to achieve computation on a nanoscale. Here, we experimentally demonstrate self-induced phase change effect which relies on the coupling between geometric changes of thin liquid film to optical properties of photonic waveguide modes, and then employ it for neuromorphic computing. In our optofluidic silicon photonics system we utilize thermocapillary-based deformation of thin liquid film capable to induce nonlinear effect which is more than one order of magnitude higher compared to the more traditional heat-based thermo-optical effect, and allowing operation as a nonlinear actuator and memory element, both residing at the same compact spatial region. The resulting dynamics allows to implement Reservoir Computing at spatial region which is approximately five orders of magnitude smaller compared to state-of-the-art experimental liquid-based systems.

Exploring novel light-matter interaction regimes is fundamental to our ability to advance our understanding of both light and matter properties and to introduce novel technological capabilities. While light-solid interaction on the nanoscale attracted prime attention due to advancement of nanofabrication methods, enabling to observe numerous nonlinear optical effects in Silicon Photonics (SiPh) and integrated photonics platforms (see refs. [1–5] and references within), exploring light-liquid interaction remains an attractive research direction due to liquids' extremely rich phenomenology and transport regimes. In particular, since liquids allow molecular transport at the cost of relatively low energy compared to thermal energy, it can support a variety of physical effects which are inherently not possible in solid systems, such as optical solitons due to the reorientation of liquid crystal molecules[6], light branching in thin liquid films[7], light-induced tuning of plasmonic resonances[8], as well as modern extreme UV laser systems[9] with applications in high-resolution lithography. One particular application which gained tremendous importance in our time is more efficient information processing allowing novel capabilities such as computer vision and object detection which were intractable through traditional von Neumann architecture approaches, spiking significant research efforts to develop new unconventional, neuromorphic computing (NC) schemes such as Recurrent Neural Networks (RNN)[10] and its subset reservoir computing (RC)[11–14]. More specifically, while originally RC emerged as an efficient computation method implemented on digital computers, realizing a physical RC platform remains an attractive possibility[15,16] because in RC paradigm the underlying physical system can perform as a reservoir and non-linearly transform the input signal such that separability via linear regression methods becomes more effective. Among various physical mechanisms, optical systems present the advantages of 'speed of light' propagation[17], high connectivity, and wavelength multiplexing[18–21], which already led to free space and on-chip realizations[22–25].

[1]Department of Electrical and Computer Engineering, University of California, San Diego, 9500 Gilman Dr., La Jolla, CA 92093, USA. [2]King Abdulaziz City for Science and Technology (KACST), P.O. Box 6086, Riyadh 11442, Saudi Arabia. ✉e-mail: rubin.shim@gmail.com

Here, we build on top of our recently theoretically proposed light-heat-liquid nonlinear-nonlocal interaction mechanism[26–28], to experimentally demonstrate the self-induced optical phase change effect which relies on an interplay between changes of liquid film's surface geometry, due to thermocapillary (TC) effect[29–32], and propagating photonic mode in silicon (Si) WG. In particular, the photonic mode partially dissipates on an integrated gold patch, thus enabling a localized heat source and TC-driven deformation of optically thin liquid film, which in turn modifies the overlap of the photonic evanescent tail with air, constituting a back-reaction profoundly affecting the phase of the optical mode. We characterize the induced phase change as a function of optical power and driving frequency of the input optical power, and then demonstrate that the information of the optical pulse magnitude can be stored in the liquid deformation for several tens of ms, allowing to perform digital XOR task as well as analog task of handwritten digits recognition from the Modified National Institute of Standards and Technology (MNIST) dataset, which are often chosen for proof of concept demonstration of nonlinear computation[18,33]. Finally, we employ NARMA2 task to numerically explore RC performance as a function of liquid cells and reservoir size in more elaborate integrated photonic circuits.

Figure 1a schematically describes the experimental setup where CW laser source of wavelength 1550 nm couples photonic TM mode into Si channel WG. The optical phase changes are detected by employing photonic Young Interferometer (YI), conceptually analogous to the classical Young double slit experiment, where the optical power is equally split between active and passive WGs. In so doing, the mode in the active WG interacts with the liquid film whereas the mode in the passive WG mode is used as a reference. Upon emission into free space the two modes form interference fringes which are monitored by Charge-Coupled Device (CCD) camera; change in liquid thickness in the active WG leads to self-induced optical phase difference and shift of interference fringes. Figure 1b presents scheme of the normal cross section where the gold patch is deposited on the top facet of the active WG allowing to dissipate light and generate surface tension gradients needed to trigger the TC effect. In the WGs we employ propagating TM mode, characterized by stronger oscillation of the electrical field along the vertical $z$ direction, as it facilitates both more efficient optical dissipation in the metal patch (deposited on top of the active WG) and higher sensitivity to changes of the liquid film thickness (compared to TE mode characterized by electric field oscillations along the in-plane direction). Figure 1c–e presents microscopy image of the photonic YI and scanning electron microscopy (SEM) of the box-shaped etched cell with the gold patch. Crucial for our ability to conduct repeatable and controlled experiments, is to deposit into the etched cell silicone oil droplets of volume of few femtoliters. Since development of femtoliter droplet deposition methods is still an active field of research[34,35], we employed triboelectric effect in order to trigger electrostatic-based silicone oil drop-by-drop emission from glass tip to silica substrate, where the emission rate is controlled by modifying the distance between the tip and the silica substrate, more details are included in Supplementary Material (SM). Figure 1f presents the preparatory step where the optical phase monotonically increases during droplet deposition process into the liquid cell; the insets describe four microscopy images of liquid's surface at corresponding moments of time. Figure 1g presents typical 3D profile of silicone oil surface, whereas Fig. 1h provides height along cell's diagonal indicating that the film tends to wet the vertical sidewalls, enabling optically thin liquid

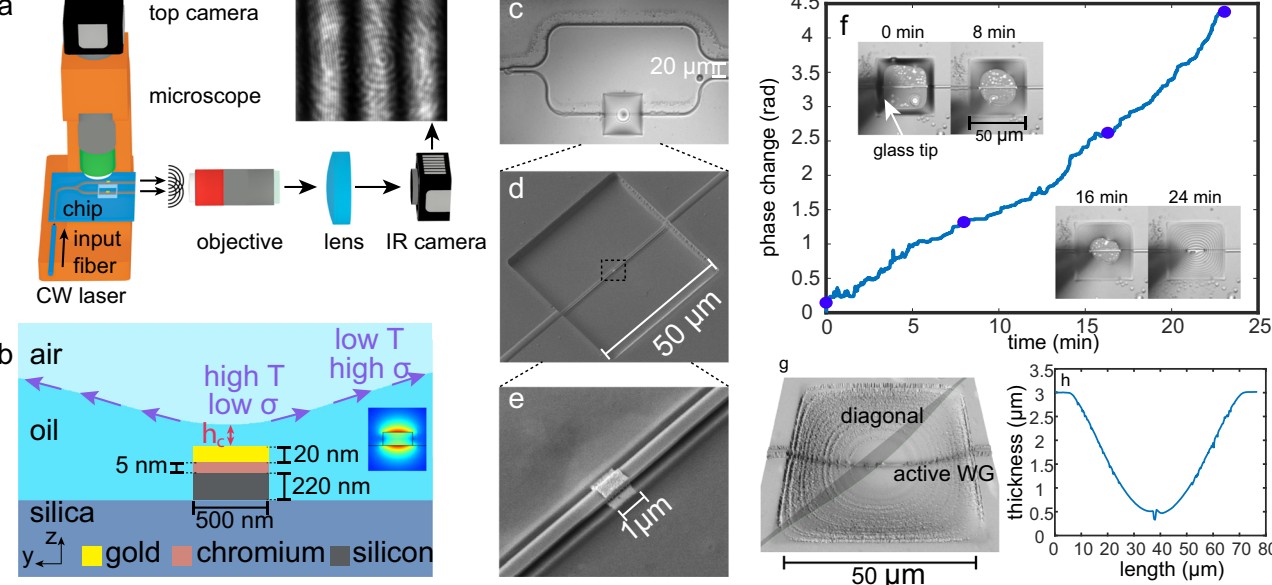

**Fig. 1 | Key components of the optofluidic system and preparatory steps allowing to observe the self-induced phase change effect and its implementation for RC. a** Schematic description of the experimental system allowing to couple continuous wave (CW) laser source of wavelength 1550 nm into SiPh chip, and detect interference fringes shift due to liquid deformation. **b** Schematic illustration of the normal section presenting 220 × 500 nm active WG, integrated 20-μm-thick gold patch (with 5-nm-thick chromium adhesion layer), as well as optically generated surface tension gradients triggering TC-driven thin liquid film deformation. The corresponding changes of the overlap of the TM mode with air leads to optical phase change and is detected by shift of interference fringes in the IR camera. **c** Top camera image of the photonic Young Interferometer (YI) circuit presenting: Y-junction, liquid cell of dimensions 50 μm × 50 μm × 3 μm hosting the active WG with liquid, and the 20 μm separated output WGs ports emitting into the

free space. **d** SEM image of the etched cell in the thermal oxide cladding used as a liquid chamber. **e** Higher magnification of the SEM image showing the metal patch on top of the WG. **f** Experimental results presenting phase difference between the two arms of photonic YI as a function of femtoliter droplet deposition process allowing to fill the etched cell with thin silicone oil film (see Methods subsection "Liquid deposition"); top camera presents the liquid cell at specified time moments with purple disks indicating the corresponding phase change values. See SM and video V1 for experimental demonstration of typical drop-by-drop deposition process. **g** 3D profile of liquid surface measured by White Light Interferometry (see Methods subsection "Optical setup and phase shift extraction"). **h** Liquid Thickness extracted from the 3D profile, indicating about 0.5-μm-thick film in the center region of the liquid cell.

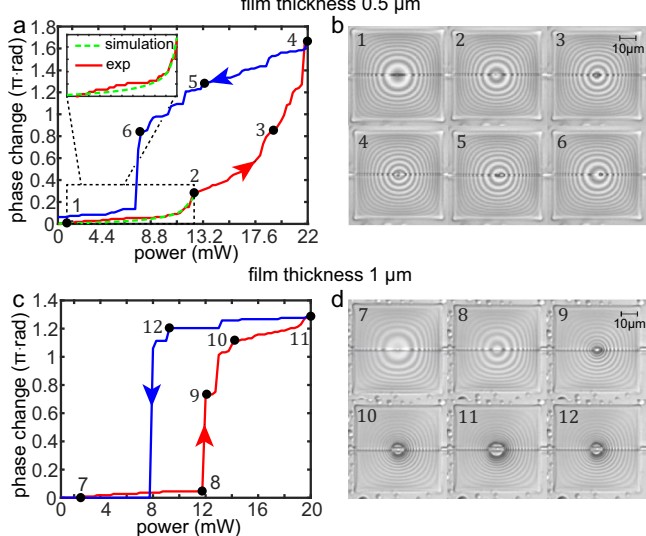

**Fig. 2 | Self-induced phase change effect as a function of incident optical power for shallow and thick liquid films, measured by utilizing photonic YI circuit. a**, **b** and **c**, **d** present experimental results of phase change as a function of the incident optical power in the active WG for 0.5 μm and 1.0 μm initial silicone oil thickness, respectively. Both cases admit gradual increase of the phase as a function of increased optical power (red curves) reaching to maximal values of $1.7\pi$ rad and $1.3\pi$ rad, and phase decrease as the power is decreased (blue curves) leading to hysteresis loop. **b** Top view optical images of the silicone oil cell corresponding to the parameters at the black points labeled between one to six on the curves presented in (**a**) and (**b**), where state 1 corresponds to initial configuration whereas state 2 to the configuration where the gas–liquid interface nearly touches the WG. Similarly, **c** presents hysteresis curve for 1-μm-thick liquid film with more abrupt transition to ruptured state. Assuming normal incidence of the red LED source, and employing the condition for destructive interference $\lambda_{LED}/2n_{oil}$, we estimate the contact angle of the rupture in configuration 11 as -5.5°. In case of a thinner film presented in (**a**) we present an inset with multiphysics simulation results showing a good comparison against experimental results. See SM and videos V2 and V3 for experimental demonstration of typical phase change effect.

film in the central region of approximate thickness 0.5 μm above the silica substrate.

## Results

### Self-induced phase change as a function of optical power

Figures 2a, b and c, d present experimental results of the self-induced phase change effect for approximately 0.5-μm and 1-μm-thick silicone oil film, respectively, as a function of laser power modified in steps of ±0.25 mW; in SM we provide an estimate of the corresponding optical power in the WG. In particular, Fig. 2a presents strictly increasing phase shift as a function of incoming optical power described by the red curve point 1-2-3-4, whereas the regime with subsequently decreased optical power, which is dominated by ambient surface tension forces and relaxation of the invoked deformation, lead to phase change values along the blue curve point 4-5-6-1 (see Supplementary Movie 2 for the entire process). Interestingly, the two curves enclose closed hysteresis loop which is typically observed in wetting experiments, and attributed to history-dependent contact angle hysteresis and also to asymmetric advancing and receding film dynamics[36,37]. For instance, the latter can stem from either topographical features or chemical nonhomogeneity factors[38,39], which are both present in our photonic chip and are represented by Si channel ridge WG on silica substrate and the gold patch.

Top view microscopy images of the silicone oil under red LED illumination (central wavelength $\lambda_{LED} = 680$ nm) of corresponding configurations presented in Fig. 2b, indicate that the transition from

state 1 to state 2 is characterized by relatively small surface deformation and does not lead to film rupture. However, for optical power above 13 mW, the liquid film evolves toward ruptured state 3 and subsequently to state 4 with increasingly larger dewetted region, which is accompanied by higher slope of the phase change curve (compared to section 1 to 3) due to more significant overlap of air with the photonic evanescent tail. After each optical intensity change, we allowed at least 3 s relaxation time in order to allow the liquid film to reach its new static configuration.

Decreasing the optical power leads to reduction of the size of the dewetted region presented in state 4 to 6 of Fig. 2b, and for power level 8 mW the liquid film experiences collapse of the ruptured region and convergence toward the initial state 1 achieved at lower power values. Note that since the same laser source is used to excite the liquid as well as to form the interference fringes, some minimal non-zero optical power is initially required to circulate in both WGs. Our finite elements numerical simulation results (green dashed curve in the lower left part of Fig. 2a)[40] present quantitative agreement against the experimental results in the pre-rupture regime 1 to 2, but is not capable of handling scenarios with ruptured film at higher optical power, due to limitation of the employed moving mesh method[28].

Figure 2c presents similar self-induced phase change effect and closed hysteresis curve for 1-μm-thick silicone oil film (see Supplementary Movie 3 for the entire process), indicating that in this case the deformation stage prior to liquid rupture between state 7 and 8 introduces very small phase change value $0.04\pi$ rad, i.e., approximately order of magnitude smaller compared to the changes in the thinner film between the pre-rupture states 1 to 2. Smaller phase change values stem from larger distance between the WG and the gas–liquid interface, and therefore to less prominent overlap of the evanescent optical mode with the gas phase. Applying higher power of 12 mW triggers liquid instability leading to abrupt change of gas–liquid interface geometry from deformed configuration 8 to ruptured configuration 9 accompanied by significant phase change values $0.8\pi$ rad. Further increase of applied optical power leads to larger values of phase change between states 9 and 11 due to dewetting of the liquid film seen in Fig. 3c. Lowering the optical power leads to phase change evolution along the blue curve and small rupture healing between states 11 and 12 Around 8 mW abrupt healing takes place accompanied by phase change of $-1.2\pi$ rad.

Importantly, while the maximal phase change reported above (Fig. 2a) is $-1.7\pi$ rad, measurement under identical optical power without silicone oil yields values below noise level indicating that the relative magnitude of TC effect relative to TO effect is at least larger by a factor of 24 (see SM S.5).

It is worth mentioning that the hysteresis effect can be associated with memory and history-dependent processes, however, as we will present in the following the memory effect we employ in this work for RC does not directly rely on hysteresis behavior, but rather on finite relaxation time of optically thin liquid film.

### Self-induced intensity change as a function of optical power and applied modulation frequency

Next we consider photonic MZI circuit with a single output port, allowing to translate phase change into intensity change and employ a point detector capable to monitor dynamical processes of liquid deformation which evolve on sub ms time scale.

First we consider DC regime, presented in Fig. 3a, where increasing input optical power leads to strictly increasing output power described by the red curve 13-14-15-16; subsequent decrease of input power leads to output power described by the blue curve 16-17-18-13 forming a closed hysteresis loop.

Contrary to the YI circuit (Fig. 2b) where fringes shift is already indicative of nonlinear response due to liquid deformation, changes of the optical power in in the MZI circuit can in principle also stem from

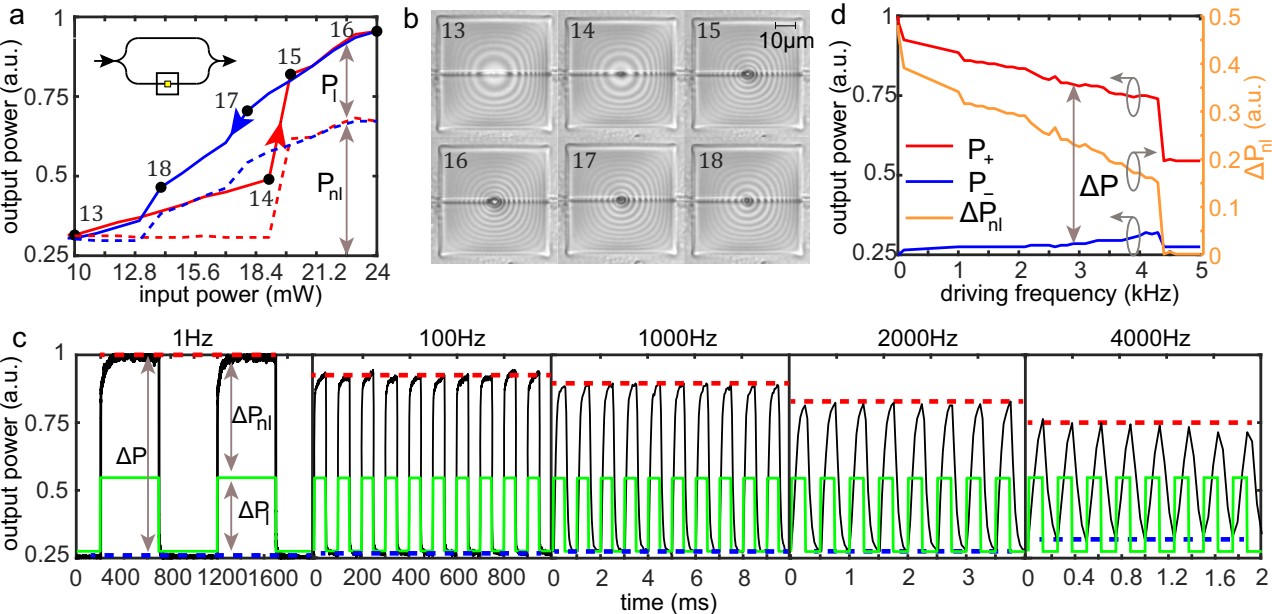

**Fig. 3 | Self-induced phase change leading to intensity modulation as a function of input power and driving frequency in MZI circuit. a** Self-induced intensity change as a function of CW optical signal, due to optically induced deformation of liquid film of initial thickness of ~1 μm. The increment steps were of ±1 mW and with relaxation time between successive power levels ~3 s to allow sufficient relaxation into stable configuration. The dashed lines ($P_{nl}$) represent nonlinear output response obtained by subtraction of the corresponding input power ($P_l$) defined by Eq. (1). **b** Top view microscopy images of the correspondingly numbered states of

the liquid film in (**a**). **c** Output signal as a function of time for representative frequencies 1 Hz, 100 Hz, 1000 Hz, 2000 Hz, and 5000 Hz where the green and black curves represent the linear and nonlinear power response, $\Delta P_l$ and $\Delta P_{nl}$, respectively. The red and blue lines represent the maximal ($P_+$) and minimal ($P_-$) values of $\Delta_{nl}$. The maximal and minimal power levels of the input signal (green curve) are 16 mW and 5.5 mW, respectively. **d** $P_+$, $P_-$, and $\Delta P_{nl}$ defined by Eq. (2) as a function of driving frequency $\omega$, which are described by red, blue, and green curves, respectively.

changes of input laser power levels which are unrelated to liquid response.

In order to discriminate between the nonlinear response due to liquid deformation and changes of input optical power, we subtract from the total measured power ($P$) the linear contribution of the input power ($P_l$), expressed as

$$P_{nl} = P - P_l, \qquad (1)$$

which isolates the nonlinear contribution. Figure 3a presents experimental measurements in photonic MZI circuit where the total power $P$ is described by solid red and blue lines whereas the nonlinear response $P_{nl}$, defined by Eq. (1), is described by blue and red dashed lines, where red/blue indicate increasing/decreasing optical power regime. In particular, at power levels between 10 and 19 mW silicone oil film (approximate thickness 1 μm) experiences relatively small deformation (pre-rupture), captured by microscopy images 13 and 14, presented in Fig. 3b, and hence leads to a negligible nonlinear response represented by a practically flat red dashed line in Fig. 3a in that power region. Nevertheless, for optical power levels above 19 mW the dashed line 14-15-16 is increasing, indicative of a nonlinear response due to liquid deformation. Similarly to the YI measurements reported in Fig. 2, subsequent decrease of the input power yields strictly decreasing curve 16−18 enclosing a closed hysteresis loop with the red curve.

Next we consider the output power of the MZI circuit under the action of AC optical input signal as a function of driving frequency ($\omega$) and liquid cell of the same thickness as in the DC case above. Figure 3c presents output power (black curve) due to square wave input optical power (green curve) for five different driving frequencies.

Similarly to our discussion before Eq. (2), we define the nonlinear response function $\Delta P_{nl}$ as

$$\Delta P_{nl} = \Delta P - \Delta P_l; \quad \Delta P \equiv P_+ - P_-, \qquad (2)$$

which eliminates changes of the input power and captures the nonlinear power response due to liquid deformation. Here, $\Delta P$ is the modulation depth of the output power, i.e., the difference between the maximal ($P_+$) and the minimal ($P_-$) values of the output power described by the red and the blue dashed curves in Fig. 3c, respectively, whereas $\Delta P_l$ is the fixed $\omega$ independent modulation depth of the input signal. Performing $\omega$ sweep over smaller steps, presented in Fig. 3d, indicates that $\Delta P$ and its nonlinear component $\Delta P_{nl}$, are both decreasing functions of $\omega$. Interestingly, $\Delta P_{nl}$ admits abrupt transition with nearly vanishing $\Delta P_{nl}$ above 4.3 kHz, indicative that the output and input signals are practically identical and hence implying that the liquid film does not respond under driving signals of sufficiently high $\omega$, explicitly demonstrated by the 5 kHz case presented in Fig. 3c. In fact, the latter could be anticipated from behavior of simpler mechanical systems such as forced damped oscillator, where the resultant amplitude decays as driving frequency tends to infinity. In SM section S.6 we present complementary measurement of the self-induced phase change effect, under fixed $\omega$ but variable film thicknesses (Supplementary Movie 5). The latter is achieved by employing electrostatic-based drop-by-drop deposition of femtoliter silicone oil droplets and reveals liquid thickness supporting maximal nonlinear response for a given square wave modulation.

## Liquid-based optical memory and basic reservoir computing

At this point we explore the possibility of utilizing the self-induced phase change effect presented above, in order to write/read optical information into/from liquid's surface and demonstrate basic RC of XOR task. To this end, we encode logical '0' and '1' as low and high power levels $P_0$ and $P_1$, respectively, where each of the pulses admits duration time $\tau_w$ and is followed by a pulse of weaker non-zero power level $P_r$ and duration $\tau_r$. The latter enables relaxation of thin liquid film toward initial non-deformed state and also to monitor fringes movement also during $\tau_r$ time intervals, which would not be possible if $P_r$

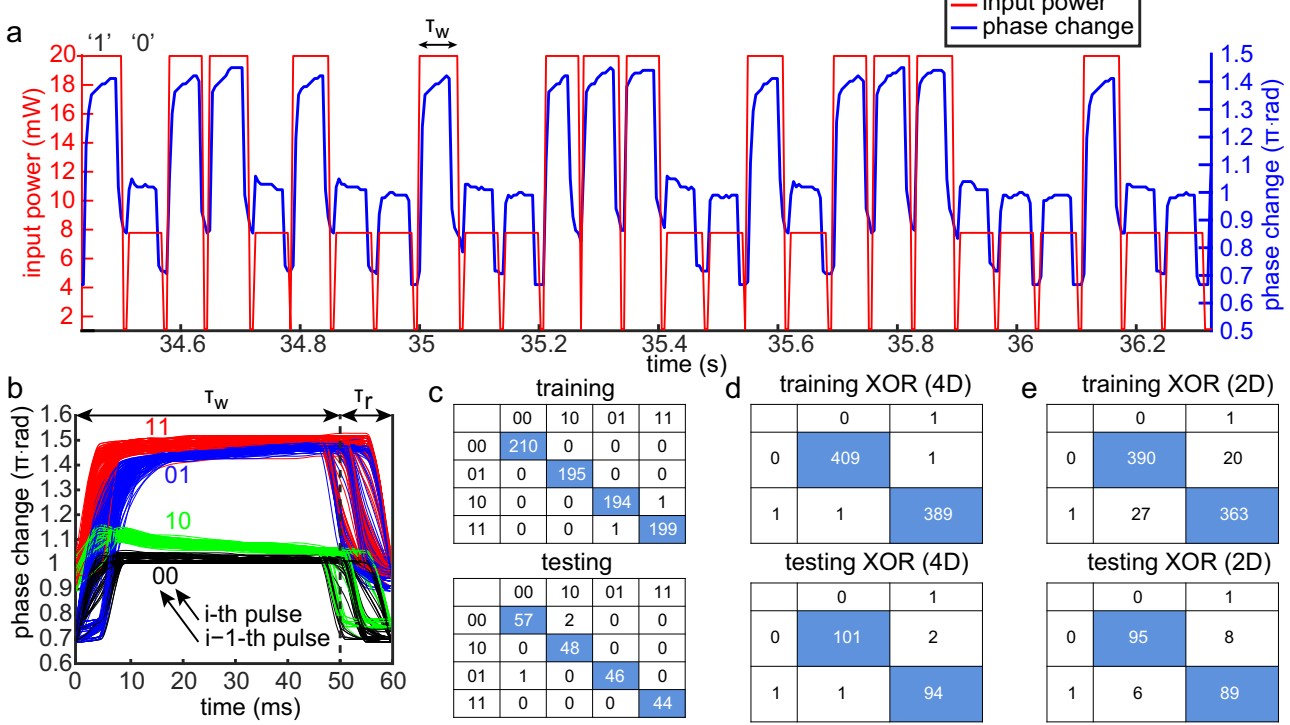

**Fig. 4 | Experimental demonstration of RC which relies on the self-induced phase change effect. a** Phase change response due to TC-driven deformation of 0.5-µm-thick silicone oil film under a sequence of square wave pulses of duration $\tau_w = 50$ ms and relaxation time $\tau_r = 10$ ms, with power levels $P_{0,1}$ encoding logical '0' and '1'. Here, the optical power levels $P_0$ and $P_1$ lead to phase change values of approximately $\pi$ rad and $1.4\pi$ rad, respectively. **b** Folded dynamics plot where all pulses of time $\tau_w$ and relaxation $\tau_r$ are plotted on top of each other along the interval of duration $\tau_w + \tau_r$ revealing four dominant clusters classified according the previous (i−1-th) and latest (i-th) pulse. **c** Confusion matrices of the 00, 01, 10, 11 classification where the vertical axis corresponds to actual (input) values whereas horizontal axis corresponds to predicted (output) values. **d, e** Confusion matrix of XOR task based on preliminary 4D and 2D classification, respectively.

would vanish. Figure 4a presents induced phase change values (blue curve) due to a random input sequence of 0's and 1's (red curve) with: $\tau_w = 50$ ms, $\tau_r = 10$ ms, $P_0 = 8$ mW, $P_1 = 20$ mW, and $P_r = 1$ mW. See Supplementary Movie 4 for the entire process. While the time response in Fig. 4a presents strong correlation between the magnitude of latest optical pulse to the value of the self-induced phase change, e.g., higher power latest optical pulse leads also to higher value of optical phase change, in practice, it can be noted even with a naked eye, that there are some patterns which are indicative of the memory carried by previous pulses in the sequence which affect the latest pulse. In order to highlight the memory effect and the correlation between the latest pulse and the pulse preceding it, we plot all time intervals of length $\tau_w + \tau_r = 60$ ms on top of each other in Fig. 4b which describe emergence of four clusters, as was recently theoretically predicted in ref. 28. Here, the clusters are labeled as '11', '10', '01', '00' where the left and right indices stand for the preceding and the latest optical pulses of power $P_0$ or $P_1$, respectively. In particular, the latest pulse '1' leads to peak which belongs to either '11' (red) or to '01' (blue) cluster, whereas preceding pulse '0' leads to either '10' (green) or to '00' (black) cluster. Note that formation of four clusters indicate that the different peaks are grouped irrespective of their location in the time-series, thus realizing echo-state property which we computationally demonstrated in the pre-ruptured regime in ref. 28.

In order to accomplish RC task we inject a random sequence of 1000 pulses with power levels $P_{0,1}$, and acquire the dynamics of the optical phase by using the CCD camera at the output of photonic YI. We then use the first 800 pulses to accomplish the training stage, digitally achieved by solving the linear ridge regression equation and obtaining the teaching matrix (see ref. 28 for details). At the next testing stage, we feed $1000 - 800 - 2 = 198$ bits, where a factor of two

describes elimination of one bit at the beginning (because XOR requires two input bits) and for convenience one bit in the end. We then acquire the corresponding dynamics as a function of time, and digitally apply the teaching matrix on the measured signal to perform the computation.

Figure 4c presents the confusion matrices for the training and the testing stages of the classification of '00', '01', '10', '11' temporal sequences (considered as mapping {00, 01, 10, 11} → {00, 01, 10, 11} and referred below as 4D case), whereas Fig. 4d presents the corresponding performance of XOR task if '00' and '11' are associated with '0', whereas '01' and '10' are associated with '1' (considered as mapping {00, 01, 10, 11} → {0, 1} and referred below as 2D case). Figure 4e, on the other hand, presents direct XOR task computation without employing the 4D space, showing less prominent performance presumably due to less effective usage of the underlying memory.

Unsurprisingly, increasing $\tau_r$ leads to more pronounced loss of memory of the preceding pulse due to more complete relaxation of the gas–liquid interface, hence resulting in test error increase presented in Fig. 5a. To quantify the notion of memory we represent each one of the self-induced phase change curves in Fig. 4b along the interval $\tau_w + \tau_r$, as a point in the principal components (PCs) space. Specifically, by keeping the two dominant PCs in the corresponding PC expansion of each one of the curves, we expect to obtain four different clusters corresponding to each one of the groups. We then enclose each cluster by corresponding standard deviation ellipsoid, indicative of variance ($\sigma$) of points' distribution along each one of the axes, and define the degree of separation between the different groups by considering intersection of the corresponding ellipses where small/large intersection indicates high/large separation due to strong/weak memory. With this in mind, assuming that the total area occupied by all

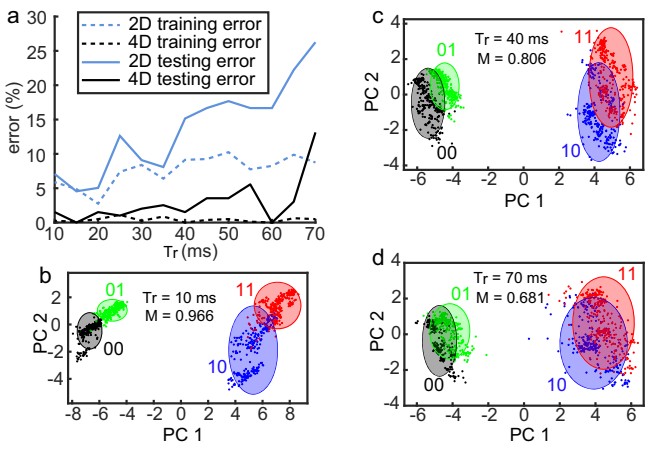

**Fig. 5 | Experimental demonstration of RC performance and memory as a function of relaxation time. a** Testing error as a function of time difference between successive pulses, $\tau_r$, varying between 10 and 70 ms at steps of 5 ms. **b–d** PCA diagrams of first two PCs (PC1-PC2 plane) for increasingly higher values of $\tau_r$: **b** 10 ms, **c** 40 ms, **d** 70 ms, indicating increasingly lower separation of the standard deviation ellipses and corresponding lower values of the memory parameter $M = 0.966, 0.806, 0.681$.

ellipses is $S_T$ we define the following non-dimensional memory parameter $M$

$$M = (S_T - S_I)/S_T, \tag{3}$$

where $S_I$ is the intersection area of different ellipses.

Figure 5b–d represents the curves as points in PC1-PC2 plane where PC1 and PC2 are the first two PCs, and the axes of the corresponding standard deviation ellipses along each direction are equal to $2\sigma$.

Since $S_I < S_T$ holds, the parameter $M$ is subject to $0 \le M \le 1$, where the limit cases $M = 0$ and $M = 1$ are realized when $S_I = 0$ and $S_I = S_T$, respectively. Notably, the ellipses in Fig. 5b–d admit increasingly higher intersection (lower separation between '11' and '10', and '01' and '00' states) for increasingly higher relaxation times $\tau_r = 10, 40, 70$ ms with corresponding values $M = 0.966, 0.806, 0.681$, respectively. While in our RC approach we employed relatively slow CCD camera, given the fact that TC-driven actuation of liquid films can support oscillations on kHz scale (see Fig. 3), it is possible to achieve faster computation if one employs line detectors which offer faster acquisition time.

After discussing the experimental realization of the digital XOR task, we move on to investigating RC-based performance of the analog task of handwritten digits recognition, using MZI setup schematically described in Fig. 3a. To this end we employ MNIST digits dataset[41], which was also employed in our previous theoretical work[28], and implement row-by-row injection of the image where the power level of the optical pulse is proportional to the brightness of the corresponding pixel, and duration of each optical pulse is 4 ms without any relaxation time in between (see "Methods" section and SM for more details). Table 1 presents classification performance of 0−1, 0−2, ... 0−9 sets where the first row stands for linear regression (LR) result, which does not rely on physical reservoir, whereas the second row presents optofluidic RC result showing lower error for all tasks (except task 0−7 of similar performance) and most significant relative error reduction for 0−1 and 0−2 cases, given by $100 \times (0.83 - 0.67)/0.83 = 19.2\%$ and $12.7\%$. Interestingly, RC efficiency of the classification task takes place at actuation pulse power $P_{max} = 13$ mW, which corresponds to a slightly ruptured liquid surface and relatively high values of self-induced phase change. However, higher $P_{max}$ values, which in turn lead to more prominent ruptured state and higher nonlinearity,

**Table 1 | Experimental result comparing RC-based analog digit recognition task of 0−1, 0−2, ... 0−9 cases, relative to the linear regression (LR) classification which does not employ physical reservoir**

|     | 0−1  | 0−2  | 0−3  | 0−4  | 0−5   | 0−6   | 0−7   | 0−8   | 0−9   |
|-----|------|------|------|------|-------|-------|-------|-------|-------|
| LR  | 0.83 | 4.33 | 8.17 | 8.20 | 12.11 | 13.10 | 14.19 | 16.17 | 18.7  |
| RC  | 0.67 | 3.78 | 7.67 | 8    | 10.28 | 13    | 14.19 | 16    | 17.75 |

All values are testing errors in %.

do not provide better classification accuracy. It may suggest that in this case the dynamics becomes chaotic, which is not conducive to achieve efficient RC, and may furthermore imply that error values of RC computation may be also used to classify the degree of chaos present in the system.

Finally, we employ NARMA2 task to investigate RC performance as a function of reservoir dimension and the number of liquid cells in more complex photonic circuits. In particular, our results indicate that using a single MZI with liquid cell in one of its arms leads to a low NMSE value 0.0023 which is lower compared to NMSE value 0.0054, obtained by applying linear regression method (see "Methods" section and SM for more details).

## Discussion

To summarize, in this work we employed chip-scale SiPh platform with a partially exposed WG, facilitating controlled light-heat-liquid interaction where the photonic mode affects the geometry of the gas−liquid interface via TC-driven liquid transport, whereas evolving shape of the liquid film surface translates into changes of the photonic mode phase. Notably, the observed optical effect takes maximal phase change values between 1.4 and $1.8\pi \cdot$ rad in the steady state regime, and $-0.65$ $\pi \cdot$ rad in the oscillatory regime, depending on initial liquid thickness, which is more than one order of magnitude higher compared to the more traditional TO effect in solid materials. Even higher phase change values are expected if higher optical power is used or more efficient optical dissipation takes place. Furthermore, our results reveal non-trivial dependence of the nonlinear phase change as a function of liquid thickness, which was achieved by employing drop-by-drop electrostatic deposition of femtoliter silicone oil droplets under fixed periodic actuation which can reach several kHz. Our 3D simulation results provide good quantitative agreement against the experimental results in the pre-rupture regime, and can further stimulate the development of computational methods to capture ruptured dynamics of the liquid film, hysteresis effects due to pinning effects and non-uniform substrate properties, and instability of the optically thin liquid films which leads to sharp changes in the self-induced phase values above threshold values. We then employ the self-induced phase change effect to demonstrate that liquid film is capable to serve as an optical memory capable to support beyond von-Neumann computational architecture where memory is a part of the computation. Remarkably, the active area of the nonlinear response due to liquid deformation takes place in a region of few μms and does not require additional feedback lines which are needed in electronic systems or in hybrid photonic-electronic systems, leading to realization of a liquid-based computational system five orders of magnitude smaller than reported in previous works[42,43]. In particular, we experimentally demonstrate the capability to perform with high accuracy digital XOR task as well as enhance the performance of MNIST digits classification analog task compared to a linear-based classifier. In future versions of our system, we expect to achieve higher accuracy on time-consuming digit classification tasks and improved performance on even more complex tasks by using mechanically more stable fiber-chip coupling scheme. The latter is expected to enable the training of the reservoir on a larger dataset and reduce the impact of random

fluctuations caused by changes in the coupling. Furthermore, providing isolation between the chip and the external environment should reduce the magnitude of external air flows which may affect the system. Finally, we employ NARMA2 task to test RC performance of more elaborate photonic integrated circuits with varying liquid cell configurations and reservoir sizes. Our results indicate that even a small number of liquid cells can significantly improve computation accuracy, indicating a non-trivial relationship between circuit structure and computation.

Our methodology and results provide a clear path to pursue versatile fundamental research directions aiming to explore intriguing optical properties such as long-range nonlocal interaction between adjacent WGs due to liquid deformation, resonant-based absorption due to excitation of localized surface plasmon polaritons (as opposed to non-resonant metal patch employed above), and also spatial and temporal properties of nematic-isotropic[44] phase change effects of liquid crystals due to local temperature increase. Providing further detailed understanding of the various factors that affect the self-induced phase change effect may lead to an extension of our work where longer memory effects enable computation applications with higher bit data. Furthermore, the connection of our optofluidic system to NC may stimulate exploration of intricate light-liquid interaction processes with computation applications, including emulation of biological neuron activity which also takes place in liquid environment.

## Methods

### Measurement principle: photonic Young interferometer

Figure 1a schematically describes the experimental setup where CW laser source of wavelength 1550 nm couples optical TM mode from polarization maintaining lensed fiber into $220 \times 500$ nm Si channel WG by employing linear inverse taper. The optical phase changes are detected by employing photonic YI circuit, conceptually analogous to Young double slit experiment, where the coupled optical mode is split by using 50−50 Y-splitter[45] into active and passive WGs. In this circuit the active WG traverses the liquid cell whereas the passive WG serves as a reference and also admits gold patch (which does not interact with liquid) in order to balance losses allowing to increase visibility of the interference fringes formed upon emission into free space.

Figure 1b presents schematic description of the normal cross section describing 1-μm-long and 20-nm-thick gold patch deposited on top facet of the active WG in order to dissipate light and operate as a localized heat source needed to trigger the TC effect. Since TM mode is characterized by stronger oscillation of the electrical field along the vertical $z$ direction, it facilitates both more efficient optical dissipation on the metal patch and higher sensitivity in thickness changes of liquid film, compared to TE mode which admits oscillations mostly in the in-plane $y$ direction. Figure 1c−e presents optical and SEM images of the photonic circuit, the etched box-shaped cell with the WG and the gold patch on top of the WG, respectively. The two output ports are set to a distance of 20 μm (see SM for details) then produce interference pattern upon emission into the free space, and the optical phase change due to liquid perturbation in the active WG are then detected by measuring shift of the fringes with CCD camera.

### Fabrication of SiPh circuit

SiPh chip was fabricated by the Applied Nanotools INC foundry. The WG patterns were defined by electron beam lithography (EBL) and reactive ion etching (RIE) processes on a Silicon-on-insulator (SOI) wafer with a 220-nm-thick Si device layer, a 2-μm-thick buried oxide layer and 725-μm-thick silicon handle wafer. Afterward, the gold patch was defined by optical maskless lithography (MPL) + magnetron sputtering lift-off and a 3-μm-thick $SiO_2$ cladding was deposited using chemical vapor deposition (CVD). In order to create windows in the oxide cladding serving as liquid cell, the pattern was first defined by MPL, and then RIE was used to etch ~2.5 μm oxide layer in order to

protect the WG structure. The remainder of 0.5-μm-thick oxide layer was removed by diluted Buffered Oxide Etch (BOE).

### Liquid deposition

Silicone oil (PHENYLMETHYLSILOXANE (cas number 9005-12-3, of refractive index 1.444), Gelest®) was deposited into the etched liquid cell by employing the triboelectric effect known to invoke electrostatic charges in dielectric materials by scrubbing glass tip (Schott Duran borosilicate glass pipette, World Precision Instruments®) with nitrile gloves. The electrostatic charges in turn led to electrohydrodynamic atomization (a.k.a. electrospray)[46] of femtoliter silicone oil droplets without the need to employ external electrodes. See Supplementary Movie 1 for a complete liquid deposition process. The distance between the tip and silica substrate was few tens of microns, and controlling this distance affected droplet emission rate from the tip. See SM Fig. S1 for droplet emission as a function of glass tip-chip distance shift, as expected indicating that higher distances lead to lower emission rate.

### Optical setup and phase shift extraction

Light properties: we employed Santec TSL-550 as a source for 1550 nm wavelength light, with maximal output power 24 mW. In order to maintain TM polarization in the WG we employed PM fiber (OZ Optics Ltd.).

Imaging: to form and capture the interference pattern (e.g., presented in Fig. 1a) we imaged the two output ports on a NIR CCD camera (Pembroke Instruments, $640 \times 512$ pixels, maximum 225 frames per second) by employing 50X objective (Mitutoyo, Plan Apo NIR) and 500 mm lens together forming image system with 125 times magnification. We then slightly shifted the objective from the back focal plane in order to form interference fringes of the two point-like sources (output ports). By adjusting the shift allows to form three fringes in the image field.

Phase change extraction: in order to extract the phase change from the acquired image, we first add all the 512 rows to obtain 1D fringe intensity distribution curve. Afterward, we employ MATLAB's built-in 'smooth' and 'findpeaks' function to smooth the curve and locate the position of the fringe. For more details, please refer to SM part 3.

Topography of silicone oil film presented in Fig. 1g was achieved by employing white light interferometry system by Profilm 3D® (Filmetrics, San Diego, CA, USA). In order to reduce index contrast we utilized silicone oil (see below) of refractive index similar to that of silica substrate.

### Multiphysics simulations

We employed COMSOL Multiphysics® software[40] in order to simulate the combined action of light propagation, fluid dynamics, heat transport and surface tension gradients, allowing to capture key trends of the self-induced phase change effect presented in Fig. 2 (see ref. 28 for more details of the simulation files). In particular, we employed computational domain of buried WG (as opposed to ridge WG employed in the experiment), allowing simpler computation and expected to yield good agreement in the pre-rupture regime. Key parameters we used to describe silicone oil according to manufacturer, include refractive index of value 1.444, as well as other parameters which slightly different from our previous work[28] and are given by: viscosity = 0.0202 Pa · s, density = 1010 kg/m³, surface tension = 0.32 N/m. We executed a batch of simulations where initially flat liquid film was subjected to TC-driven deformation due to increasingly higher values of in-WG power, varying from 0.5 mW to 3 mW in steps of 0.1 mW. The value of 3 mW was chosen as slightly lower compared to the 22/6 mW = 3.6 mW where 22 mW is the maximal optical power of the source presented in Fig. 2, whereas the factor of 1/6 corresponds to power reduction when coupled to chip (see SM S.4). According to our simulation results, in-WG

power levels above 2.6 mW (corresponding to laser source power $2.6 \times 6 = 15.6$ mW), lead to liquid dynamics evolving toward the bottom of the liquid cell, whereas power levels below 2.6 mW lead to stable configurations (for initial liquid thickness 0.5 µm). Here, stable configuration of liquid film is characterized by negligible changes of its thickness over time period of a few msec.

As some of the key parameters values, such as silicone oil's Marangoni constant, and the actual value in-WG optical power, are not exactly known to us, complete comparison to the experimental results is precluded at this point. Nevertheless, we note that the ratio of the in-WG optical power for which rupture occurs in the simulation ($p_r^{(th)} = 15.6$ mW) and in the experiment ($p_r^{exp} = 13$ mW), is close to one ($p_r^{(exp)}/p_r^{(th)} = 0.83$). Hence, assuming that the in-WG optical power is $6 \cdot p_r^{(exp)}/p_r^{(th)} = 4.98$ times smaller (instead of factor 6 mentioned SM S.4), would imply that the theoretical and experimental values of rupture power agree. Under this assumption the computational results in Fig. 2a (green dashed curve), is very close to the experimental curve both admitting concave property. Taking into account additional factors such as different values of Marangoni constant and initial curvature of the deposited liquid film are expected to contribute to deviations from the computational model, and studying those effects is beyond the scope of this work.

### Optofluidic memory analysis

To obtain the principal components of the dynamical patterns we employed MATLAB's built-in 'pca' function and considered the first two principal components. Afterward, we constructed standard deviation elliptical regions and employed Mathematica to compute the intersection areas. While employing higher number of principal components will lead to higher dimensional ellipsoidal regions and likely to more accurate definition of optofluidic memory, this direction is beyond the scope of this work. We also include sample of our Matlab code importing raw experimental data and demonstrating RC via the link[47].

### MNIST task

For training and testing, we employed 3000 MNIST images, where each pixel was encoded as a 4 ms pulse with power proportion to pixel's brightness. In our experiments, we down-sampled the image from $28 \times 28$ to $14 \times 14$ to ensure that fiber-chip coupling does not change during the computation time. In SM we bring a representative signal encoding row of one of the images as well as the reservoir's response. Furthermore, we bring the confusion matrix for the full 0–9 classification test.

### NARMA2 task

The number of time steps used for training and testing of NARMA2 task is 400 and 100, respectively. The architecture of MZIs follows design proposed in ref. 48, but in our case also incorporates liquid cells in all or some MZIs arms. The numerical simulation was performed by using MATLAB[49] where the corresponding code can be found in ref. 50.

## Data availability

The raw and processed data that support the findings of this study are available from https://github.com/gckkkk/Optofluidic-RC.git and Supplementary Materials.

## Code availability

Computer codes for experimental data analyze are available from https://github.com/gckkkk/Optofluidic-RC.git. Computer codes for NARMA2 numerical simulation are available from https://github.com/pgaur7/Reservoir-Computing.git.

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

## Acknowledgements

The authors cordially thank Defense Advanced Research Projects Agency (DARPA), Nature as Computer (NAC) program management team for stimulating discussions during the program and to Prof. Maxwell Monteiro's inputs in Reservoir Computing part. DA would like to thank King Abdulaziz City for Science and Technology (KACST) for the support during his study. This work was supported by the Defense Advanced Research Projects Agency (DARPA) DSO's NAC (HR00112090009) and NLM Programs, the Office of Naval Research (ONR), the National Science Foundation (NSF) grants CBET-1704085, NSF ECCS-180789, NSF ECCS-190184, NSF ECCS-2023730, NSF ECCS-2217453, the Army Research Office (ARO), the San Diego Nanotechnology Infrastructure (SDNI) supported by the NSF National Nanotechnology Coordinated Infrastructure (grant ECCS-2025752), the Quantum Materials for Energy Efficient Neuromorphic Computing - an Energy Frontier Research Center funded by the U.S. Department of Energy (DOE) Office of Science, Basic Energy Sciences under award #DE-SC0019273, and the ASML/Cymer Corporation.

## Author contributions

S.R. conceived the project and designed the research with C.G. and Y.F. C.G. designed and taped out the sample. C.G. designed and performed the experiments. P.G. did the numerical simulations. C.G. did the Multiphysics simulation. C.G., S.R., and P.G. analyzed the data. D.A. assisted in fabrication. S.R. and C.G. wrote the manuscript. Y.F. supervised the project.

## Competing interests

The authors declare no competing interests.
