## [Peer Review File · Nature Communications]

REVIEWER COMMENTS

Reviewer #1 (Remarks to the Author):

The authors present a novel optofluidic SiPh nonlinear system which can be used for Reservoir Computing (RC). The thermocapillary effect, i.e. liquid deformation and nonlinear phase-shift generated by optical input power, was utilized as the nonlinear mechanism. This mechanism incorporates strong nonlinearity (an order of magnitude stronger than thermo-optic effect is claimed) and it includes nonlinear memory. The authors use the proposed system to demonstrate successfully the XOR task in an RC system.

The paper is well written and the proposed system, albeit slow, is quite interesting. I only have minor comments/corrections to add.

Minor revisions/comments:

- In the introduction, the "speed of light" is claimed as an advantage of using optical systems. Unless justified, this statement is meaningless and should be removed.

- Although the nonlinear memory is verified to be in the order of tens of milliseconds, in the RC task presented here it was only utilized for 2 bits. What about tasks that require longer memory (more than 2 bits)?

- How was only TM ensured at the input of the chip? The laser must have both TE and TM modes excited. Please add this information.

- It is mentioned that the refractive index of the Si oil was similar to that of silica. What was the exact value of n_{oil} assumed in the simulations?

- Fig.3(c): Could you please include traces for >4.3 kHz as well? Also, there is an error in the label: "yellow" should be "green".

- There is a typo in the text line before Eq.(2): "he nonlinear" should be "the nonlinear".

- It is mentioned that the output is "resembling spiking neuron". Please remove this statement as it is incorrect and there is no evidence to back up such a claim.

- Regarding the "2D" XOR task more information is required. Does the "2D" refers to the labeling for the output of the training? E.g. 4D: {00,01,10,11} -> {00,01,10,11}, 2D: {00,01,10,11} -> {0, 1}?

- I believe more elaboration is needed regarding the memory factor M. Why a smaller intersection S_I signifies a larger M? What is the physical meaning of S_I ?

- In the discussion section it is claimed that "almost 2π " could be achieved for DC state. However, based on the results presented this is an overstatement. In Fig. 2 only $1.2\pi \sim 1.6\pi$ is shown, which is not "almost 2π ". Similarly, "nearly π " is claimed for the AC state. However, based on the results presented in Fig. 4(b) that is another overstatement. Please correct this and mention the actual achieved values in your claims.

- What is "NC" in the discussions?

Reviewer #2 (Remarks to the Author):

This paper describes a device that uses multiphysics of fluid and light to achieve nonlinearity (tentatively called an OFSPM (Optofluidic self-phase modulator) in this review) and its application to reservoir computing (RC). OFSPMs are very interesting and, as described in the paper, have great potential for sensing and other applications that solid-state devices cannot provide.

However, since the configuration of OFSPM and the XOR operation as an RC have been described in detail in the literature [27], there seems to be a lack of new content for publication in this journal. Certainly, the use of observation of Young interference systems in the output part is new, though it would be nice to see new functions or dramatic improvement in performance by using this configuration, or ideas for large-scale computation as an RC. In particular, I could not read from this paper whether the proposed RC could function as a reservoir to store a lot of information because it would relax quickly. Even if this system could have sufficient capacity, hysteresis may inhibit the Echo State Property (or consistency) generally required for RCs, and therefore, a more detailed study seems necessary. In addition, as a new point, the paper evaluated the hysteresis of OFSPM. However, the title

of the paper gives the impression that it is about optofluidic memory for RC, but there seems to be a contradiction between the title and the content because the hysteresis of OFSPM, as described in the paper, is not used for RC.

Other than that, my suggestions for improvement and questions are as follows:

1) Maybe I am misunderstanding, but in figure 4a, b, the system has strong linearity and does not appear to be capable of XOR operation. An example of time series pattern with XOR operation would help the reader to understand.

2) Is the XOR operation appropriate as a benchmark task for the reservoir computing? Wouldn't the experiment be like tuning the system to reproduce XOR as nonlinear dynamics?

3) I could not find the definition of "4D and 2D classification" in Figure 4.

Reviewer #1 (Remarks to the Author):

The authors present a novel optofluidic SiPh nonlinear system which can be used for Reservoir Computing (RC). The thermocapillary effect, i.e. liquid deformation and nonlinear phase-shift generated by optical input power, was utilized as the nonlinear mechanism. This mechanism incorporates strong nonlinearity (an order of magnitude stronger than thermo-optic effect is claimed) and it includes nonlinear memory. The authors use the proposed system to demonstrate successfully the XOR task in an RC system.

The paper is well written and the proposed system, albeit slow, is quite interesting. I only have minor comments/corrections to add.

Minor revisions/comments:

- In the introduction, the "speed of light" is claimed as an advantage of using optical systems. Unless justified, this statement is meaningless and should be removed.

We agree with the reviewer and now clarify that the "speed of light" advantage of optical systems for computation resides on the fast propagation speed of optical fields. In fact, the latter never constitutes a bottleneck in terms of carrying information in both active and passive systems, irrespective of the corresponding material platform. To highlight this point, we have now added the following text in the introduction, with a new reference [17].

Among various physical mechanisms, optical systems present the advantages of the so-called 'speed of light' propagation, especially prominent in passive optical systems [17], which rely on one time pulse pass and hence do not rely on light modulation. Furthermore, light propagation is never a limiting factor also in systems imposing nonlinear optical modulation due to dynamics of relevant material degrees of freedom. Other advantages of optical fields include high connectivity, and wavelength multiplexing [18–21], which already led to free space and on-chip realizations [22–25].

- Although the nonlinear memory is verified to be in the order of tens of milliseconds, in the RC task presented here it was only utilized for 2 bits. What about tasks that require longer memory (more than 2 bits)?

We agree with the reviewer and would like to clarify that in this work we rely on our previous theoretical paper (ref [28] in the revised manuscript), where we numerically demonstrated that relying on memory which resides beyond the preceding bit is less efficient for RC (Fig.5f in [28]). Therefore, while in principle employing "longer memory" is possible, from an experimental perspective it would require more sensitive instrumentation and a better understanding of the factors which contribute to the magnitude of the self-induced phase change. Furthermore, considering more than two bits is an interesting extension of the current setup, which may also be realized by relying on longer memory or multiple intensity levels which again would require more sensitive measuring equipment and hence beyond the scope of this work.

To highlight this point, we have now added the following text in the discussion:

"Providing further detailed understanding of the various factors that affect the self-induced phase change effect may lead to an extension of our work where longer memory effects enable computation applications with higher bit data."

–How was only TM ensured at the input of the chip? The laser must have both TE and TM modes excited. Please add this information.

TM polarization was ensured by employing a laser source whose output is polarization maintaining (PM) fiber, where the latter also acts as a polarizer. We have now explicitly mentioned this detail in the text:

In order to maintain TM polarization in the WG we employed PM fiber (OZ Optics Ltd).

–It is mentioned that the refractive index of the Si oil was similar to that of silica. What was the exact value of n_{oil} assumed in the simulations?

We now explicitly mention in Liquid Deposition and Multiphysics sections that the refractive index of the liquid used in the simulations is 1.444, which coincides with the value reported by manufacturer of the silicone oil we used, and also coincides with the value of fused silica at relevant wavelength.

Key parameters we used to describe silicone oil according to manufacturer, include refractive index of value 1.444, as well as other parameters which are slightly different from our previous work and are given by:...

–Fig.3(c): Could you please include traces for >4.3 kHz as well?

We thank the reviewer for this comment and agree that it is important to include the data for the regime above 4.3 kHz where liquid deformation does not follow optical excitation. The rightmost graph in Fig.3c includes now data corresponding to 5 kHz and is accompanied by the following text:

Interestingly, dP_{rii} admits abrupt transition with nearly vanishing dP_{rii} above 4.3 kHz, indicative that the output and input signals are practically identical and hence implying that the liquid film does not respond under driving signals of sufficiently high ω , explicitly demonstrated by the 5 kHz case presented in Fig.3c

- Also, there is an error in the label: "yellow" should be "green".

We fixed the error in the label of Fig.3c.

- There is a typo in the text line before Eq.(2): "he nonlinear" should be "the nonlinear".

Typo fixed.

- It is mentioned that the output is "resembling spiking neuron". Please remove this statement as it is incorrect and there is no evidence to back up such a claim.

We agree with the reviewer and have removed the statement because indeed in our manuscript we have never discussed a formal proof that the emerging dynamics satisfies conditions of spiking neurons.

- Regarding the "2D" XOR task more information is required. Does the "2D" refers to the labeling for the output of the training? E.g. 4D: {00,01,10,11} -> {00,01,10,11}, 2D: {00,01,10,11} -> {0, 1}?

We now clarify on p.11 that indeed 2D and 4D schemes correspond to the following labeling schemes {00,01,10,11} -, {0,1}, {{00,01,10,11} -, {00,01,10,11}}, respectively, and added the following text:

Fig.4c presents the confusion matrices for the training and the testing stages of the classification of '00', '01', '10', '11' temporal sequences (considered as mapping {00,01,10,11} -, {00,01,10,11} and referred below as 4D case), whereas Fig.4d presents the corresponding performance of XOR task if '00' and '11' are associated with '0', whereas '01' and '10' are associated with '1' (considered by mapping {00,01,10,11} -, {0,1} and referred below as 2D case).

- I believe more elaboration is needed regarding the memory factor M. Why a smaller intersection S_I signifies a larger M? What is the physical meaning of S_I?

We agree with the reviewer that more details regarding the emerging memory factor $M = (S - S_I) / S_{\square T \square}$, where $S_{\square T \square}$ and S_I stand for the total area and the intersection area of the standard deviation ellipses in the Principal Component (PC) space, respectively, described in Fig.5. First, we would like to clarify that the four (blue, red, black, green) types of curves described in Fig.4b can be classified not only according to the latest pulse, but also according to the preceding pulse which is indicative of memory effect in the system. In the extreme case of weak memory, the red and blue curves would collapse into a single group and similarly the green and the black curves would collapse into a different group. Therefore, the degree of separation between the blue-red and green-black curves is indicative of the degree of memory in the system where high separation (low intersection) is indicative of strong memory and vice versa low separation (high intersection) is indicative of weak memory. Next, we note that in order to parametrize the degree of separation between the red and blue curves as well as the green and black curves we employ PC description of each one of the curves in Fig.4b, where each curve is described by the first two dominant PCs, leading to four clusters in the PC1-PC2 plane. It is then natural to enclose each cluster by standard deviation ellipses and define the degree of separation between the different groups by considering intersection of the corresponding ellipses where according to the above-mentioned considerations small/large intersection

indicates strong/weak memory. In particular, smaller intersection means it is easier to distinguish between the two signals in the PC space and vice versa with higher intersection.

To clarify this point, we have now added the following text:

"To quantify the notion of memory we represent each one of the self-induced phase change curves in Fig.4b along the interval $T_{WW} + T_r$ as a point in the principal components (PCs) space. Specifically, by keeping the two dominant PCs in the corresponding PC expansion of each one of the curves, we expect to obtain four different clusters corresponding to each one of the groups. We then enclose each cluster by corresponding standard deviation ellipsoid, indicative of variance of points' distribution along each one of the axes, and define the degree of separation between the different groups by considering intersection of the corresponding ellipses where small/large intersection indicates high/large separation due to strong/weak memory. With this in mind, assuming that the total area occupied by all ellipses is S_T we define the following non-dimensional memory parameter M

$$M = (S_T - S_I) / S_T,$$

where S_I is the intersection area of different ellipses."

- In the discussion section it is claimed that "almost 2pi" could be achieved for DC state. However, based on the results presented this is an overstatement. In Fig. 2 only 1.2pi--1.6pi is shown, which is not "almost 2pi". Similarly, "nearly pi" is claimed for the AC state. However, based on the results presented in Fig. 4(b) that is another overstatement. Please correct this and mention the actual achieved values in your claims.

We agree with the reviewer and provide now more accurate values in the discussion, nevertheless higher phase change could be in principle achieved if higher laser power is employed. To reflect this point, we now write:

Notably, the observed optical effect takes maximal phase change values between 1.4 - 1.8 $it \cdot rad$, in the steady state regime, and approximately 0.65 $it \cdot rad$ in the oscillatory regime, depending on initial liquid thickness, which is more than one order of magnitude higher compared to the more traditional TO effect in solid materials. Even higher phase change values are expected if higher optical power is used or more efficient optical dissipation takes place.

- What is "NC" in the discussions?

We have now defined the acronym NC (Neuromorphic Computing) in the introduction.

Reviewer #2 (Remarks to the Author):

This paper describes a device that uses multiphysics of fluid and light to achieve nonlinearity (tentatively called an OFSPM (Optofluidic self-phase modulator) in this review) and its application to reservoir computing (RC). OFSPMs are very interesting and, as described in the paper, have great potential for sensing and other applications that solid-state devices cannot provide.

We thank the reviewer for finding our work as “very interesting” and admitting “great potential”.

However, since the configuration of OFSPM and the XOR operation as an RC have been described in detail in the literature [27], there seems to be a lack of new content for publication in this journal.

We thank the reviewer for this comment and would like to clarify that our previous work (referred as [28] in the revised manuscript), is theoretical/computational, whereas the present work is experimental with theory only supporting a small part of the steady state self-induced deformation in the pre-rupture regime. Therefore, it is not clear how designing and characterizing dedicated optofluidic chip-scale setup, developing novel dedicated electrostatic deposition method and using these to report for the first time about experimental observation of the self-induced phase change effect, memory effects in liquid film on submicron scale and then experimentally demonstrate RC, could be interpreted as “lack of new content.” In fact, for centuries the scientific method as we know it today was built around experiments validating theoretical models/theories or about experimental observations and then building theories to explain them, where unvalidated theoretical models never qualify as valid physical paradigm.

Certainly, the use of observation of Young interference systems in the output part is new, though it would be nice to see new functions or dramatic improvement in performance by using this configuration, or ideas for large-scale computation as an RC.

We would like to clarify that since this manuscript is the first to use Young Interferometry on-chip to study nonlinearity of liquid films and in fact also the first to employ geometrical changes of liquid film surface to produce optical nonlinearities, most of our results revolve around the more fundamental themes such as AC regime, novel drop-by-drop deposition (exclusively developed for our system), and the memory effect. We then demonstrate that one could in principle use the type of memory manifest in our system to perform RC of nonlinear XOR task. Interestingly, while in the theoretical paper [28] we considered several different physical effects such as self-induced phase change, self-induced coupling change and self-induced transmission, in the present experimental manuscript we focus exclusively on the self-induced phase change effect and most of the aspects we cover (besides pre-rupture steady state regime Fig.2a) do not admit straightforward extension of our already extensive computational capabilities and hence is not covered by [28].

Indeed, our work admits direct extensions by constructing more waveguides which could in principle increase throughput and perform more complex RC tasks, however we feel that it then would overshadow the more fundamental findings and would result in too long and convoluted paper.

In particular, I could not read from this paper whether the proposed RC could function as a reservoir to store a lot of information because it would relax quickly. Even if this system could have sufficient

capacity, hysteresis may inhibit the Echo State Property (or consistency) generally required for RCs, and therefore, a more detailed study seems necessary.

We thank the reviewer for the comment and would like to clarify that:

1. Following RC paradigm (e.g., references [15,18,19] in the revised manuscript) it is not needed to store large volumes of information or whole sequences in the reservoir in order to perform RC. Instead, a necessary condition for a given physical system to qualify as a reservoir is if it admits short-term memory needed to mix the latest pulse with the preceding one. In so doing it realizes the so-called echo state property (ESP) where the response does not depend on the initial conditions but on the preceding and the latest pulses. In our system short-term memory is naturally realized by liquid's property to deform and sustain the deformation for few tens of ms and in our previous work ([28], Fig.S6a) we demonstrated how the system "forgets" about initial conditions. However, in many other systems (e.g., photonic-electronic) this relaxation time is even shorter by few orders of magnitude than in our system (see for more detailed comparison table S2 in [28]), yet those systems (e.g., [18]) still realized experimental RC because ESP condition is satisfied.

2. Echo state property: Our experimental results presented in Fig.4b clearly indicate that the pulses are grouped into four clusters depending on the power values of the latest and the preceding pulses, irrespective of their location within the injected sequence (presented in Fig.4a). Hence, these results serve as an empirical proof that our system satisfies ESP and hence it is not surprising that the computation results presented in Fig.4 present very good performance. It is also worth mentioning that based on pre-rupture regime, we expected our system to be able to serve as RC as in our previous work [28] we derived to the following understanding:

"More formally, since the dynamics of our system converge to states that do not depend on the initial conditions, it satisfies the so-called common-signal-induced synchronization (see Ref. 38 and references within), also known as echo state, which is a necessary property enabling a dynamical system to serve as an RC."

In our experimental work, we validated this result and extended it to a regime which was not covered in our simulations where the liquid film is ruptured.

To reflect this point, we have now added:

"Note that formation of four clusters indicate that the different pulses are grouped irrespective of their location in the time-series, thus realizing echo-state property which we computationally demonstrated in the pre-ruptured regime in [28]."

3. Hysteresis: Our experimental plots presented in Fig.2 indicate that in our system there is a hysteresis which arises when we apply increasingly higher values of optical power and then gradually decrease it. However, it is different from the experimental work point used to derive the experimental results in Fig.4 where optical power admits either low or high powers levels, which are injected without gradual power changes in between. Hence, it is not clear how to directly relate hysteresis curves in Fig.2 to RC computational results in Fig.4. Furthermore, formation of the four clusters in Fig.4a is consistent with the intuitive understanding presented above supporting the nearly perfect experimental RC results, and also consistent with the simulation results presented in previous work [28] (valid in the more limited pre-rupture

regime). Nevertheless, studying how hysteresis and the underlying pinning effects due to surface non-homogeneity and other factors may affect the evolution of the liquid film and the corresponding computation is a legitimate research direction.

To reflect this point, we now added in the discussion:

“Our 3D simulation results provide good quantitative agreement against the experimental results in the pre-rupture regime, and can further stimulate the development of computational methods to capture ruptured dynamics of the liquid film, hysteresis effects due to pinning effects and non-uniform substrate properties, and instability of the optically thin liquid films which leads to sharp changes in the self-induced phase values above threshold values.”

In addition, as a new point, the paper evaluated the hysteresis of OFSPM. However, the title of the paper gives the impression that it is about optofluidic memory for RC, but there seems to be a contradiction between the title and the content because the hysteresis of OFSPM, as described in the paper, is not used for RC.

We thank the reviewer and would like to clarify that in our work we have experimentally demonstrated that we can optically “write” the information about the magnitude of the optical pulse power. This information can be stored if the liquid film did not relax, and subsequently can be optically “read” by the subsequent optical pulse. This is markedly different from the hysteresis type of behavior which we report in Fig.2 and Fig.3a, where using a series of increasing values of incident optical power which are followed by series of decreasing power levels leads to history dependent behavior presumably due to pinning effects and surface non homogeneity. Our results presented in Fig.4b presenting four distinctive cluster, where belonging of each curve to a particular cluster is determined by the latest and the preceding pulses. Furthermore, nearly perfect performance of optofluidic RC, indicate that at least in the current experimental system we have no evidence that hysteresis leads to long term memory effects. Therefore, the dominant memory mechanism we used for RC relies on finite time relaxation of the liquid film (leading to self-induced phase change effect), and hence we naturally chose to mention both in the corresponding title.

To summarize, Fig.2 and Fig.3a present the basic response of the self-induced phase change as a function of increasing and subsequently decreasing power levels, and both indicate the presence of hysteresis behavior. However, we never claimed to use hysteresis as memory element for RC. To emphasize this point we now mention in the text:

“It is worth mentioning that the hysteresis effect can be associated with memory and history dependent processes, however, as we will present in the following the memory effect we employ in this work for RC does not directly rely on hysteresis behavior, but rather on finite relaxation time of optically thin liquid film.”

Other than that, my suggestions for improvement and questions are as follows:

1) Maybe I am misunderstanding, but in figure 4a, b, the system has strong linearity and does not appear to be capable of XOR operation. An example of time series pattern with XOR operation would help the reader to understand.

We would like to clarify that while the optical response, to a large extent, is indeed correlated with the magnitude of latest optical pulse, i.e., higher power latest optical pulse leads to higher value of optical phase change, in practice since we are working in the regime where the temporal separation between subsequent optical pulses is smaller than the relaxation time of the liquid film, there is “nonlinear” contribution of the preceding pulse to the phase change generated by the subsequent (latest) optical pulse. Specifically, Fig.4b describes formation of four clusters which are separated according to the latest pulse **and also** according to the preceding pulse. In our analysis we take advantage of this non-trivial memory property in order to perform computation of XOR operation presented in Fig.4c-e.

In order to highlight this point, we have now added the following sentence on the bottom of p.10:

While the time response in Fig.4a presents strong correlation between the magnitude of latest optical pulse to the value of the self-induced phase change, e.g., higher power latest optical pulse leads also to higher value of optical phase change, in practice it can be noted even with a naked eye, that there are some patterns which are indicative of the memory carried by previous pulses in the sequence which affects the latest pulse.

2) Is the XOR operation appropriate as a benchmark task for the reservoir computing? Wouldn't the experiment be like tuning the system to reproduce XOR as nonlinear dynamics?

Yes, XOR operation is commonly employed as a benchmark test for RC (as pointed out also in the reference [18]). However, in RC one does not need to worry about specific output numerical values *per se* as in analog computing, but rather more concerned if the dynamics of the physical system satisfies proper conditions which include [12]: (i) *separation property*: different inputs should be mapped onto different reservoir states; (ii) *approximation property*: reservoir states that are only slightly different should be mapped onto identical targets; and (iii) *fading memory*: previous series values are stored and mixed with the input and usually, only recent inputs are relevant while those from the far past do not need to be considered (as in delayed XOR task we considered and also in other applications such as speech recognition). This was generated by mixing the latest pulse with the preceding pulse which requires some amount of memory. Typically, for a given physical RC system with pre-determined dynamics one can tune the input signal to achieve better memory and optimal performance. In our case, we tune the value of the relaxation time τ_{rr} and learn from Fig.5a that the optimal performance of XOR task takes place for τ_{rr} around 20 ms.

3) I could not find the definition of "4D and 2D classification" in Figure 4.

We thank the reviewer for this comment and have now added on p.11 definition for 4D and 2D cases.

REVIEWER COMMENTS

Reviewer #1 (Remarks to the Author):

The authors have addressed all reviewers' comments adequately. I have nothing further to add.

Reviewer #2 (Remarks to the Author):

The authors have responded to all comments from reviewers in the revised paper and the "Response to Referees Letter," and the paper seems to be getting clearer.

However, it seems important to clarify the relationship between the performance of the reservoir computer and the characteristics of the system, since the implementation of reservoir computing is a dedicated device, not a natural computational medium. In this paper, the XOR operation is used as a benchmark problem, but it is so simple that there is a concern that the characteristic problem is not visible. If possible, more complex benchmarks such as the NARMA10 task, which is the benchmark problem for many reservoirs, should be used to show the characteristics. Alternatively, it seems that the relationship between the performance of the reservoir computer and the system parameters of the system proposed in this paper and its scalability should be described. Clarification of these points would make this paper suitable for publication in Nature Communications.

Second response to reviewers (manuscript NCOMMS-22-47625-T)

Reviewer #1 (Remarks to the Author):

The authors have addressed all reviewers' comments adequately. I have nothing further to add.

We thank the reviewer for the previous valuable comments and suggestions which improved the quality of our work, and for finding our work appropriate for publication in the journal Nature Communication.

Reviewer #2 (Remarks to the Author):

The authors have responded to all comments from reviewers in the revised paper and the "Response to Referees Letter," and the paper seems to be getting clearer.

We thank the reviewer for the previous constructive comment and were satisfied to learn that the manuscript is now clearer.

However, it seems important to clarify the relationship between the performance of the reservoir computer and the characteristics of the system, since the implementation of reservoir computing is a dedicated device, not a natural computational medium. In this paper, the XOR operation is used as a benchmark problem, but it is so simple that there is a concern that the characteristic problem is not visible. If possible, more complex benchmarks such as the NARMA10 task, which is the benchmark problem for many reservoirs, should be used to show the characteristics.

We thank the reviewer for raising this point and agree that a more complex computational benchmark enables better visibility of reservoir performance, its properties and furthermore allows to suggest a relationship between information processing and the physical behavior of optofluidic system. Consequently, we experimentally employed handwritten 0-9 digits classification (from MNIST database) in order to explore the performance of the proposed optofluidic reservoir and learn its limitations within the framework of a particular hardware. Furthermore, we employed NARMA2 task, due to the prominent short-term memory present in our system, in order to theoretically assess its computing performance in more complex integrated photonic circuits as a

function of number of input channels and liquid cells. Depending on the physical structure of the liquid cell and the optical encoding of the data, the memory could be increased for higher-order NARMA benchmark tests, but a study on the optimization of RC for a particular memory length is beyond the scope of this work.

To highlight these findings the manuscript now includes the following additional material **Page 13-15**

After discussing the experimental realization of the digital XOR task, we move on to investigating RC-based performance of the analog task of handwritten digits recognition using MZI setup schematically described in Fig.3a. To this end we employ MNIST digits data set [18], which was also employed in our previous theoretical work [30], and implement row-by-row injection of the image where the power level of the optical pulse is proportional to the brightness of the corresponding pixel and duration of each optical pulses is 4 ms without any relaxation time in between (see Methods section and SM for more details). Table.1 presents classification performance of 0 – 1, 0 – 2, ... 0 – 9 sets where the first row stands for linear regression (LR) result, which does not rely on physical reservoir, whereas the second row presents optofluidic RC result showing lower error for all tasks (except task 0 – 7 of similar performance), and most significant relative error reduction for 0 – 1 and 0 – 2 cases, given by $100 \times (0.83 - 0.67)/0.83 = 19.2\%$ and 12.7% .

	0-1	0-2	0-3	0-4	0-5	0-6	0-7	0-8	0-9
LR	0.83	4.33	8.17	8.20	12.11	13.10	14.19	16.17	18.7
RC	0.67	3.78	7.67	8	10.28	13	14.19	16	17.75

TABLE 1. Experimental result comparing RC-based analog digit recognition task of 0 – 1, 0 – 2, ... 0 – 9 cases, relative to the linear regression (LR) classification which does not employ physical reservoir. All values are testing errors in %.

Interestingly, RC efficiency of the classification task takes place at actuation pulse power $P_{max} = 13$ mW, which corresponds to a slightly ruptured liquid surface and high values of self-induced phase change. However, higher P_{max} values, which in turn leads to more prominent ruptured state and higher nonlinearity, does not provide better classification accuracy. It may suggest that in this case, the dynamics becomes chaotic which is not conducive to efficient RC, and may furthermore imply that error values of RC computation may be also used to classify the degree of chaos present in the system. Finally, we employ NARMA2 task to investigate RC performance as a function of reservoir dimension and the number of liquid cells in more complex photonic circuits. In particular, our results indicate that using a single MZI with liquid cell in one of its arms leads to a low NMSE

value 0.0023 which is lower compared to NMSE value 0.0054, obtained by applying linear regression method (see methods section and SM for more details).

Page 16

In particular, we experimentally demonstrate the capability to perform with high accuracy digital XOR task as well as enhance the performance of MNIST digits classification analog task compared to a linear-based classifier. In future versions of our system, we expect to achieve higher accuracy on time-consuming digit classification tasks and improved performance on even more complex tasks by using mechanically more stable fiber-chip coupling scheme. The latter is expected to enable the training of the reservoir on a larger dataset and reduce the impact of random fluctuations caused by changes in the coupling. Furthermore, providing isolation between the chip and the external environment should reduce the magnitude of external air flows which may affect the system. Finally, we employ NARMA2 task to test RC performance of more elaborate photonic integrated circuits with varying liquid cell configurations and reservoir sizes. Our results indicate that even a small number of liquid cells can significantly improve computation accuracy, indicating a non-trivial relationship between circuit structure and computation.

Page 20

For training and testing, we employed 3000 MNIST images, where each pixel was encoded as a 4 ms pulse with power proportion to pixel's brightness. In our experiments, we down-sampled the image from 28×28 to 14×14 to ensure that fiber-chip coupling does not change during the computation time. In SM we bring a representative signal encoding row of one of the images as well as the reservoir's response. Furthermore, we bring the confusion matrix for the full 0 – 9 classification test.

Page 21

The number of time steps used for training and testing of NARMA2 task is 400 and 100, respectively. The architecture of MZIs follows design proposed in [62], but in our case also incorporates liquid cells in all or some MZIs arms. The numerical simulation was performed by using MATLAB [50] where the corresponding code can be found in [63].

Page 32

Fig.S7 presents numerical simulation result of RC of NARMA2 task as a function of number of liquid cells and the number of inputs/outputs (reservoir size) in the photonic network. The number

of time steps used for training is 400 whereas for testing is 100 throughout the considered cases with maximal power 0.1 mW. Fig.S7a presents RC results with an MZI hosting a single liquid cell in one of the arms, demonstrating very good agreement between the predicted and the actual signals with average NMSE value 0.0023. Fig.S7b presents a schematic description of photonic

FIG. S7. Numerical results demonstrating RC of NARMA2 task using optofluidic circuits as a function of number of inputs/outputs as well as number of liquid cells. **a** Comparison between RC result and the actual values demonstrating low NMSE value 0.0023. **b** Typical architecture of photonic circuits used to study NARMA2 performance as a function of reservoir size and number of liquid cells. **c** NMSE as a function of reservoir size (number of input/output WGs), demonstrating improvement compared to the case without reservoir (labeled as '0'). In all cases only single liquid cell is used. **d** NMSE as a function of number of liquid cells in a network with a fixed number of eight inputs/outputs, and placing liquid cells in random position.

photonic WGs network where each intersection is a symmetric directional coupler, and each rectangle is a liquid cell. In such architecture the number of layers is equal to the number of inputs/outputs (reservoir size), and facilitates signal mixing from all input WGs prior to arrival to the output layer. Fig.S7c presents NMSE as a function of reservoir size due to RC performed in circuits described in Fig.S7b (for which the reservoir size is 8). In particular, reservoir size 0

indicates linear regression-based performance without utilizing the reservoir, hence indicating that employing reservoir size 2 already improves the performance by a factor of ~ 2.5 , whereas naively increasing the reservoir size does not lead to significant improvement for the specified values. The latter suggests that further optimization could be made, and in fact Fig.S7d presents NMSE values as a function of number of liquid cells in a reservoir presented Fig.S7b, where the liquid cells are positioned randomly across the photonic circuit. Interestingly, increasing the number of cells beyond 8 does not significantly improve the performance. It is worth mentioning that in the studied optofluidic system the most dominant short-term memory is one step back in time, and hence computation accuracy reduces as a function NARMA order. For completeness, NMSE values of testing stage for NARMA2, NARMA3 and NARMA 4 tasks are 0.0023, 0.0485 and 0.0596, respectively, whereas the corresponding NMSE values obtained by implementing linear regression only are given by 0.0054, 0.063 and 0.0595.

Page 34

Fig.S8 presents experimental results of handwritten 0 – 9 digits recognition (MNIST data set) using MZI with a single input and a liquid cell embedded in one of its arms, and a single output. In our experiments we down sampled the image from 28×28 to 14×14 to ensure that fiber-chip coupling does not change during the computation time (see Fig.S8a for representative images). Fig.S8b a typical encoded signal injected to the reservoir, implementing row by row image injection, as well as response signal after it passed through the reservoir. The corresponding confusion matrix summarizing RC classification of handwritten 0 – 9 digits is given in Fig.S8c naturally indicating better performance for more distinct digits (e.g., 0 and 1) and high error values (encoded as % in the colorbar) for less distinct digits (e.g., 4 and 9).

Alternatively, it seems that the relationship between the performance of the reservoir computer and the system parameters of the system proposed in this paper and its scalability should be described. Clarification of these points would make this paper suitable for publication in Nature Communications.

We agree with the reviewer and thus performed a study aiming to understand the effect of scalability on the performance of the reservoir. Specifically, we first sweep the size of the reservoir by increasing the number of input and output WGs (Fig.S7b), with increasingly large number of MZIs in the reservoir, and keeping a single liquid cell through all simulations. Size 0 reservoir indicates linear regression computation operating directly on the input without employing any reservoir. These results are summarized in the following Fig.S7c.

To evaluate the effect of the number of liquid cells in the reservoir, we consider a reservoir with 8 input and 8 output WGs, and 28 MZIs, where each MZI can in principle accommodate one liquid cell in one of the arms, implying that the total number of available slots for liquid cells is 28. Sweeping the number of liquid cells in the circuits by step of 4, and placing them in random positions we obtain the following Fig.S7d, indicating that even using 8 out of 28 available slots leads to significant enhancement of NARMA2 computation.

REVIEWERS' COMMENTS

Reviewer #4 (Remarks to the Author):

The authors have carefully considered the reviewer's comments and incorporated their suggestions into the paper. They have also clarified the research position and included prospects to enhance reader understanding. Due to these improvements, I recommend publishing this paper as an article in Nature Communications.

Third response to reviewers (manuscript NCOMMS-22-47625-T)

Reviewer #4 (Remarks to the Author):

The authors have carefully considered the reviewer's comments and incorporated their suggestions into the paper. They have also clarified the research position and included prospects to enhance reader understanding. Due to these improvements, I recommend publishing this paper as an article in Nature Communications.

We thank the reviewer for careful reading, stimulating suggestions and recommending publication of our manuscript in the journal Nature Communication.